# Assessment of Attitudes Toward Physical Education by the Implementation of an Extracurricular Program for Obese Children

**DOI:** 10.3390/ijerph17155300

**Published:** 2020-07-23

**Authors:** Ena Monserrat Romero-Pérez, Oscar Núñez Enríquez, Gabriel Gastélum-Cuadras, Mario Alberto Horta-Gim, Jerónimo J González-Bernal, José Antonio de Paz

**Affiliations:** 1Division of Biological Sciences and Health, University of Sonora, 83000 Hermosillo, Sonora, Mexico; ena.romero@unison.mx (E.M.R.-P.); mariohgim@gmail.com (M.A.H.-G.); japazf@unileon.es (J.A.d.P.); 2Faculty of Physical Culture Sciences, Autonomous University of Chihuahua, 31000 Chihuahua, Chih, Mexico; gastelum@uach.mx; 3Department of Health Sciences, University of Burgos, 09001 Burgos, Spain; 4Institute of Biomedicine, University of León, 24071 León, Spain

**Keywords:** childhood obesity, attitudes, moderate and vigorous activity, physical education

## Abstract

The World Health Organization (WHO) identifies the importance of implementing physical activity programs such as physical education (PE) classes in schools. This study identifies the attitudes of obese children toward PE, before and after participation in a vigorous-intensity physical exercise program without the participation of normal-weight peers using a questionnaire on Attitudes toward Physical Education (CAEF). 98 children between 8–11 years of age were randomized in an Experimental Group (GE) (*n* = 48) and a Control Group (CG) (*n* = 47). They were assessed using a questionnaire on Attitudes toward Physical Education (CAEF). All the study participants exhibited a BMI Z-score ≥ 2. Before the intervention, the only difference between boys and girls was “empathy to teacher and physical education subject” (*p* = 0.001, d de Cohen = 0.72, r = 0.34). The interaction between gender and training was only present in empathy for the teacher, with a medium effect size (*η*^2^ = 0.055). The implementation of PE with two hours per week elicits only a few effects over the attitude of obese children, even though with a certain engagement of gender through training in the adjustment of empathy for teachers and the PE class.

## 1. Introduction

Obesity has been defined as an increase in body weight resulting from excess fat, which in turn, jeopardizes health significantly [1]. It is a multifactorial disease caused by social, physiological, metabolic, genetic, and psychologic factors; excess food intake and reduced caloric output combine to increase the incidence in children, which rises every day [2]. Among all the health-related and psychological factors affected by obesity, it has been discussed that children and youth present a lack of energy and low self-esteem [3]. The National Health Survey in 2016 confirmed that Mexico has been documenting a growing tendency toward overweight and obesity in school children under the age of 11 years [4].

According to Weigley [5], the Body Mass Index (BMI) is a reasonable estimate of the accumulated fat in the pediatric population. It is, however, difficult to establish cut-off points for diagnosis in children since they undergo a constant height and weight evolution; therefore, for diagnosis, the comparative reference are children of the same age and gender, using percentile tables [6]. This confirms the diagnostic criteria for overweight as any percentile above 85 and obesity as any percentile above 95 [7]. The BMI z-score is also employed in the diagnosis of obesity in children [8] (number of standard deviations above or below the average value of BMI in regards to the group median of the same age and gender). The BMI z-score relates to the percentile, depending on the reference population; the 95 percentile usually corresponds to a BMI z-score of 1.68 [9]. However, the cut-off points used by the WHO for obesity in children are a BMI z-score of 2 [10], which has been used to define obesity in children in the study sample.

Several approaches have been used in Mexico as preventive measures according to the Mexican Official Norm [NOM] [11], which in its suggestions includes the design of low-calorie diet programs along with high-intensity physical activity (PA) as per the needs identified in a community. As such, it appears that the natural scenario, ideal for the application of designed protocols regarding obesity turns out to be the school setting, (by nature the place where a child spends a considerable number of hours a day interacting with a large number of peers) specifically in PE class [12,13,14,15]. This is because there is clear evidence presenting that enforced programs in the schools exert a positive effect on the conduct and healthy lifestyles [16,17,18].

Since its inclusion in the school curriculums, PE has been strongly linked to health [13]. Moreover, this association allowed an understanding of the health benefits and also helped recognize the barriers faced by children and youth engaging in PA settings like physical education. In Mexico, elementary education is provided from first grade (6 years old) to sixth grade (11 years old). Studies demonstrate this stage to be of utmost importance as the acquisition of most healthy habits occurs at this stage [19,20,21]. PE class turns out to be a favorable environment for the enforcement of these habits within the school schedule [22]. It is important to mention that the PE teacher should be one of the main agents of change to facilitate the adoption of healthy habits. The pedagogic approach in these settings must encourage and demonstrate a promising attitude as a way to acquire healthy habits. However, in most cases, a good number of PE classes do not have such emphasis [23].

According to Escarti [24], PE class must be a tool for facilitating instead of suppressing behavior. This is consistent with Kirk’s [25] work, who mentioned that a boring and unexciting environment forms a barrier for children and youth in PE. This means a PE setting should be a space where personal and collective agreements must be made for the achievement of healthy habits within school premises. Thus, having an impact on attitudes that create benefits in the development of school children at a psychological level, in their lifestyles, and their emotional and social aspects is a pre-requisite [17].

In a PE class, obese students may face specific issues that do not represent a definite personality disorder, although, such children more often show a higher rate of psychological barriers in comparison to children with normal weight [26]. Children with obesity tend to show feelings of low self-worth and self-limitations facing social isolation, and stigmatization; sometimes they also develop a feeling that the PE class is a hostile environment where they are victims of bullying and disrespect, thus creating negativity in respect to their self-esteem and body image [3,27,28]. This situation can either be cared for or worsened by the perception of physical education classes these students have, as in case of an inadequate balance between the cooperative learning between teachers and students.

The objectives of the present study include identification of the attitudes of obese boys and girls toward PE class; an analysis of modifications in attitudes of obese boys and girls toward the PE class after participating in a vigorous-intensity physical exercise program without normal-weight peers; and in case of changes, to determine if these changes are the same in boys and girls.

## 2. Material and Methods

A total of 104 school children were grouped in an experimental and a control group separately, both with children with obesity. The *Questionnaire for attitudes toward physical education* (CAEF) was used to screen the study participants for a 20-week pre- and post-intervention period [29]. This questionnaire was applied in both groups before and after of intervention.

### 2.1. Sample

The study participants comprised a total of 104 school children from 3 different elementary schools in the city of Hermosillo, Sonora, and Mexico, aged 8 to 11 years. However, only 95 participants completed the study. The participants had a BMI z-score > 2 and were randomly assigned to either an experimental group (*n* = 48) and a control group (*n* = 47) (boys and girls separately), using a computer-generated block randomization sequence (block sizes of 4) (as shown in Table 1). An open invitation was given from the research group to the principals of the schools. As a way to raise awareness on the topic, the open invitation included one presentation per school (three in total) highlighting the benefits of exercise and the risks involved in being overweight and obese during school age. The experimental group was engaged in an exercise program. It was suggested to the control group to engage during the intervention in their regular daily activities; however, the group was offered to enroll in the same exercise program once the 2-week evaluation was complete. The inclusion criteria necessitated the diagnosis of clinical obesity according to the criteria of the WHO at the time of the study, based on a standard deviation of BMI z-score > 2. The exclusion criteria were based on the presence of every single type of disability in the participants that deterred them from performing any physical activities and the presence of chronic conditions like hypothyroidism and uncontrolled juvenile diabetes Type 1. Informed consent was obtained from both the parents and the children. The Bioethics Committee of the University of Sonora supported the study (Reg. DMCS/CBIDMCS/D21).

### 2.2. Exercise Program

The exercise program was implemented 2 times per week for 20 weeks. A total of 40 sessions for 60 min each were completed and involved varied aerobic and strength activities that worked on conditional and coordination skills, physical displacements with different length and intensity, throws of balls with hands and legs, jumps, and physical activities with the opposition without complete pause between the different activities, at an average intensity during the session with a 79.8% of their maximum heart rate, (estimated by the formula, HRmax = 220-age), which was monitored with a POLAR Model FT7 heart rate monitor (Seppo Säynäjäkangas, Kempele, Filand). These sessions were held in addition to the regular PE classes already established in their school program and only children with obesity participated in the activities of this program.

The exercise program was conducted in the participant’s school. The exercises were taught by the first author and researcher, who is not their regular PE teacher. The program was implemented on separate days such that it did not interfere with their weekly mandatory PE class.

### 2.3. Instruments

The participants were weighed on a Model 803 digital SECA scale (seca gmbh & co. kg, Hamburg, Germany), with a maximum capacity of 150 kg and a sensitivity of d = 0.1 kg. For measurement of height, a Seca^®^ 213 (seca gmbh & co. kg, Hamburg, Germany) measuring rod with a maximum height of 2 m (6.5 feet) was used. The BMI was calculated using the data thus obtained according to the following formula: Weight/size^2^ (kg/m^2^). BMI z-score based on age and gender was calculated using the WHO software version 3.2., 2011 (WHO and UNICEF, Genève, Switzerland) [30].

### 2.4. Physical Capacity

The physical capacity was assessed by evaluating Curl Up, Shoulder stretch, Push up, Trunk lift, performed using the Fitnessgram methodology [31], along with horizontal jump feet together with a run time of 400 m(s).

### 2.5. Attitudes toward Physical Education

Questionnaire for attitudes toward physical education [CAEF] [29]: It consists of 56 items wherein the students are enquired about their degree of satisfaction; they can answer on a scale of 1 to 4 where 1 denotes total disagreement and 4 signifies total agreement. This questionnaire has been applied in different studies [32,33,34,35]. Puhl and Heuer [35] mention that children with obesity tend to be bullied in PE class, which in turn affects their future engagement in a healthy lifestyle. Contreras et al. [32] mention similarities stating that peer interaction directly affects their personality development. However, there is no literature about the use of this instrument in a Mexican PE context as yet.

The instrument considers the range of attitudes toward PE in 7 hypothetical factors that are related to the corresponding items for each point.

Assessment of the most essential aspect of the PE, i.e., faculty and subject.Subject acceptance in terms of its comparison with other subjects.The validity of the subject and its contents for the integral formation of the student body.Concerns of PE faculty toward the students.Performance of PE subject.Attitude toward PE and sports.Comparison between PE and sports.

### 2.6. Statistical Analysis

Statistical analysis was performed with SPSS v.23.0 (SPSS Inc., IBM, Armonk, NY, USA). Data are expressed as mean ± standard deviation. The Kolmogorov–Smirnov test was performed to test the normality assumption. Student’s independent t-tests were used to examine differences between groups and Student’s paired t-tests for pre-post means within each group, and Cohen’s d was used to measure the effect size. The main outcome measure, the modification of attitude of obese boys and girls toward PE classes, was analyzed with the two-way ANOVA (sex and intervention group), interaction analysis partial eta squared (*η*^2^) as a measure of effect size. A level of significance was established at *p* < 0.05.

## 3. Results

Table 2 depicts the values of participants related to age, weight, size, BMI, BMI z-score, and abdominal perimeter. However, no differences were observed in the baseline values between boys and girls.

Table 3, shows the results of the physical condition tests, before and after the intervention. In EG changes are observed in Curl Up, Shoulder strech and 400 m. In the remaining tests, there were significant changes in both groups.

Figure 1 shows the characteristic elements of the PA program concerning the participating children’s average heart rate throughout the 40 sessions for exercise during 20 weeks.

Table 4 presents the data collected from the different items on the CAEF questionnaire for boys and girls in the baseline; the only difference between them was in the item relating to “the empathy for the teacher and the PE subject”. This factor contains items that describe the children’s ability to establish a good relationship with their PE teacher by identifying a number of his/her traits such as the degree of concern shown toward his/her students, the PE teacher is more “fun” than other teachers, or the student has a better relationship with the PE teacher than with the rest of the teachers.

Table 5 illustrates the values for different items of the questionnaire in all the groups, both before and after the period of intervention. Girls from the GC show a significant difference in the pre and post results on their assessment of the PE subject and teacher, whereas boys from the GC show empathy for the PE teacher and subject.

Table 5 shows the values for the different factors of the questionnaire in all groups, both before and after the intervention period, and the interaction effect of intervention and gender. The organization and concordance results are unchanged and very similar for the control groups; for the experimental groups, both girls and boys showed a decrease in the initial score.

In the difficulty factor, the scores were interpreted in reverse, and it was observed that girls from both the groups (GE and GC) showed higher levels than boys; so for them, the PE class was not difficult; however, after the exercise, the values decreased for girls in the GE, which can be interpreted as an increased perception for the difficulty of the class.

Regarding the empathy factor, the girls in GC did not reflect variations in their scores over time, while the girls in the GE, on the contrary, showed a decrease in results after participating in the PE program, i.e., they reported lowered empathy. Children in both groups acquired quite similar values and modified the teacher’s perception.

In the factor that evaluated physical education as a sport, the participants in the GC showed no difference in these concepts. For girls in the GE, conceptualization about EF class and sport was interchangeable after exercise, whereas boys in this group had similar views but with higher scores, not establishing a clear differentiation between the two activities.

In the preference for PE and Sport in GC vs. GE, participants in GE indicated a decrease in their preference for this subject after the exercise. Boys in the GC had lower values than girls, and the values did not change in post-major. On the other hand, despite having higher values at the beginning of the intervention, participants from the GE showed a tendency of the degree of preference for the PE and sport to worsen after the intervention, significantly in girls.

The usefulness of PE on evaluation reflected the little worth perceived by children from both the groups. However, in children from the GE, a change was reflected after the exercise that was not quite significant, where possible for that group of children; PE is an important subject the usefulness of which would be reflected in the future.

After the intervention period, girls and boys from the GE showed significant changes in their assessment of the PE class. This aspect grouped those test items in which students assigned either more or less importance, variations in degrees of satisfaction with their PE class, whether they consider that the knowledge they are receiving is necessary and important, how fair and unbiased they feel the teacher’s evaluations are, the motivation provided by the teacher, the use of educational materials, and the equal treatment shown by the teacher for boys and girls.

Concerning the subject’s degree of difficulty and the preference for PE and sports, the girls from the GE showed significant differences after the training session. This difficulty was in comparison with other subjects that the children studied every day, and it also involved items related to how easy the children thought it was to pass the course in comparison with other subjects. Their values were interpreted in reverse; therefore, after the moderate to vigorous training, the girls from the GE expressed that the PE class had a higher degree of difficulty after the added training sessions.

Regarding the preference shown by boys and girls toward PE class, significant changes were observed in the girls’ GE after the training since they now preferred to be physically active than to be with friends or watch TV. The remaining factors assessed did not show any differences between the boys and girls as far as gender is concerned.

On analyzing the interaction between gender and training, there was no interaction, except on the empathy for the PE teacher and subject. However, since the effect size (partial eta squared) is considered small when the partial eta squared value is ≤0.02, medium when the value is ≤0.06, and large when the value is ≤0.26, it should be emphasized that even though this interaction is significant, it is also medium and by itself does not explain nine percent of the differences.

Table 5 demonstrates the evaluations of physical tests carried out. Improvements were observed in both the groups in which the lumbar force, the articular amplitude of the shoulder, and the upper and lower extremities were evaluated along with the abdominal force. Although the greatest improvements (delta: post-pre value) were presented in the EG, in the run time of 400 m, the abdominal and lumbar force, as well as a greater decrease in the sum of the cutaneous folds, was noted.

## 4. Discussion

The improvement in physical fitness was not one of the main objectives of the study, although the PE has been observed to have brought changes in some aspects of physical fitness. Some of these changes could be attributed to the growth and maturation of these children throughout the 20 weeks of the intervention as the GC also showed improvements in the manifestations of upper and lower extremity strength, lumbar strength, shoulder joint width, and decrease in the summation of skin folds. These changes occur widely in the manifestations of physical condition throughout growth [36].

However, the group of children in whom exercise sessions were implemented twice a week showed a higher gain in abdominal strength, 400-m performance, and a greater decrease in subcutaneous fat. Nevertheless, the main results of this research differ from those of Moreno, Rodriguez, and Gutierrez [29] and Cook-Cottone, Case, and Feeley [37] where boys showed a clear preference for attending PE classes in comparison to girls. This is similar to Breslin et al. [38] where they mention that PA is pleasing to both genders; however, vigorous activities requiring a considerable effort are not appealing to girls.

### Pre-Post Intragroup Questionnaire

To know the degree to which the differences found between the pre-and-post values of the group that participated in the intervention are attributable to the exercise program, intragroup comparisons were made. On analyzing variations in attitudes toward PE as assessed by the CAEF Questionnaire, it was observed that the consistency with which the teacher organized the class, perception of the teacher’s attire, the time of taking classes, and the fact that the program lacked greater practicality for the group of children show a significant change (*p* < 0.05), with a significant size of the effect on boys (Cohen’s d-0.72, r-0.52) and girls (Cohen’s d-0.50, r-0.25) essentially, 52% and 25% of pre-post differences in boys and girls, respectively, were explained by exercise intervention.

The perception of the degree of difficulty in the PE class was evaluated on the level of effort that represented to accredit the contents of the subject in comparison to other classes. It was inversely graded and the results showed a significant change (*p* < 0.05) before and after the intervention in the girls of the GE and the size of the effect (Cohen’s d-0.61, r-0.38). This points out that 38% of these changes could have been developed by the physical exercise program.

The factor related to the degree of preference for PE and sport (relating to the conceptualization that the student has about these two ideas, which sometimes are understood as synonyms) showed how the girls of GE had significant changes (*p* < 0.05) with an effect size (Cohen’s d = 73, r = 54) that determines how 54% of these changes are attributable to the intervention.

The usefulness of physical education was evaluated through items that questioned the validity of contents during the integral training of the student and were obtained from answers such as “Physical education is boring”, “what I learn in physical education is useless”, etc. It is evaluated in reverse so that high scores on the scale such as those shown by this study, reflect the little usefulness perceived by children of both the groups in this subject. Only the girls in the GC exhibited a significant change (*p* < 0.05) and showed a size of the effect (Cohen’s d-0.12, r-0.02), and hence, only 2% of these changes should be attributed to the passage of time since they did not participate in the exercise program.

PE elements such as sport and PE utility are factors that showed no significant changes before and after the program, regardless of gender.

The factor empathy reflects the teacher’s ability to engage with the students, considering the teacher “more fun” or the “teacher with whom they relate better than the rest”.

Regarding the attitude before the PE class, only differences in empathy toward the teacher were observed (*p =* 0.001) between the boys and girls, with a small effect size (Cohen’s d = 0.72, r *=* 0.34) before the intervention. This means that only 34% of the differences are explained by gender. Regarding the perceived difficulty in PE class, even if not significant, a change in the GE after participating in the PE program was noted; a similar result was obtained with the factor relating to the preference for PE and sports in the trained GE. It is important to mention that difficulty in the perception of the PE class is a factor that children consider to engage and increase their performance [39]. The only factor showing significant changes in the boys and girls from the GE before and after a vigorous exercise program is the one related to empathy for the PE teacher and subject; these results are in accordance with those obtained by Mowatt, DePaw, and Hulac [40].

However, the parameters of evaluation on the part of the students have changed, showing the importance of the teacher’s performance for this purpose in the process of coordinated learning as well as the perception of support in certain psychological needs such as autonomy and social relations and also the role that the teacher plays as a facilitator for these [41].

The recent years have seen a renewed appreciation for the importance of PE class in school settings; the PE teachers have sought specialization and professionalization of their practice, and therefore, the students’ appreciation for the subject holds immense importance. The same was established by O’Brien, Hunter, and Banks [42] clearly showing that the main objective of PE classes is to generate positive attitudes and interest toward it.

The literature now reports strong scientific evidence on the importance of promoting PA programs [17,43,44]. The present research also includes designing programs promoting PE among obese children, which has helped focus on school settings where PE classes are carried out [43]. This suggests, that PE is an ideal place to utilize an appropriate curriculum that is attractive to children, in this case with obesity, promoting engagement in PA for a lifetime [25,44]. It also encourages the analysis of different ways of delivering the planned content, implementing an approach that is not only based on the teacher–student relationship [45,46] but one that promotes a surrounding where all students feel comfortable sharing their abilities and possibilities to create an appropriate pedagogical environment that would help them become active for a lifetime.

Undoubtedly, it would be of great interest to analyze the level of motivation that students possess during the class, in subsequent studies, considering this aspect as a factor that generates a positive attitude and favors participation, especially when it comes to overweight and obese populations.

It is pertinent to highlight the importance of the role played by the PE teacher in promoting positive experiences within the class, which guarantees adherence to the development and maintenance of physical-sports habits [47,48]. This work possesses some limitations, the first one being reduced intervention time, while lack of sensitiveness in the measuring instrument is the second one as insufficient sensitivity of the instrument in detecting change could prove to be a major drawback. Perhaps, adding an interview process would help understand the attitudes toward physical education class in a deeper sense. Besides, the use of heart rate monitoring may not be the best way to quantify the intensity of exercise in children.

## 5. Conclusions

This paper presents answers to two questions raised by researchers. In regards to whether the training generates change, it can be proved that implementing PE classes through an additional two-hour weekly exercise program of vigorous intensity for obese children only produces little effects on the students’ attitude toward PE. This effect is regardless of gender, even though there is a clear gender interaction in the training, as the only change observed was in the empathy for the teacher and the PE class. However, it is essential to point out that certain limitations such as the location of the school and school-time form barriers for further exploration of different possibilities among obese children in terms of implementation of extra hours every week.

However, this also helped understand that the PE teacher and/or facilitator must use empathy to his/her favor and their good relationship with the students to increase their sensitivity and knowledge. Utilization of an attractive curriculum where the teacher/facilitator designs activities that motivate and encourage students’ participation in intense physical exercises to establish a program with motivating strategies that would promote more participation of obese children can best be implemented by PE classes.

## Figures and Tables

**Figure 1 ijerph-17-05300-f001:**
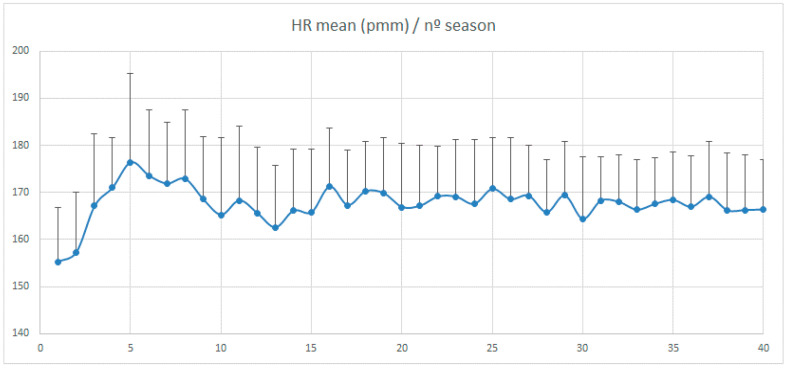
Mean heart rate per session.

**Table 1 ijerph-17-05300-t001:** Depiction of the sample according to age and group.

	Control Group (CG)		Experimental Group (EG)	
Age (year)	8–9	10–11	8–9	10–11
(*n*)	25	22	25	23

**Table 2 ijerph-17-05300-t002:** Age and anthropometric characteristics of girls and boys in the sample (Mean and SD).

Variable	Girls	Boys	*p*
Age (year)	10.3(0.8)	10.4(0.9)	0.684
Weight (kg)	53.4(11.6)	54.0(9.6)	0.284
Height (m)	139.5(9.1)	142.1(7.6)	0.138
BMI (kg/m^2^)	27.2(3.3)	26.6(3.1)	0.408
Zscore BMI	2.9(0.5)	3.1(0.8)	***0.031***
Waist–hip Index	89.4(8.0)	88.8(8.6)	0.712

(Bold and italic = statistically significant).

**Table 3 ijerph-17-05300-t003:** Evaluation of the physical condition before and after the intervention and the pre-post differences.

Test	CG				EG				
	Baseline	Post	*p*	Delta	Baseline	Post	*p*	Delta	*p* Delta
Curl Up (*n*)	8.1(3.9)	8.6(4.2)	0.206	0.5(2.3)	7.0(5.7)	8.5(5.9) *	***0.034***	2.0(0.3) #	***0.000***
Shoulder strech (cm)	4.9(4.0)	5.6(3.9)	0.282	0.7(1.5)	5.1(7.9)	6.1(6.9) *	***0.040***	1.2(2.7)	0.078
Push_up (s)	3.9(2.0)	5.2(1.7) *	***0.000***	1.4(1.5)	2.8(2.9)	3.5(2.3) *	***0.035***	0.6(2.1)	0.083
400m (s)	127(34.1)	129.8(3.9)	0.255	2.6(17.3)	111.3(26.5)	106.0(26.2) *	***0.034***	−5.4(12.8) #	***0.002***
Trunk_lift(cm)	32.7(5.2)	34.8(5.0) *	***0.003***	2.1(1.9)	32.8(6.3)	34.2(6.2) *	***0.044***	1.5(2.5)	0.063
H.jump feet together(m)	0.97(0.16)	0.99(0.17) *	***0.048***	0.02 (0.05)	0.97(0.17)	1.00(0.16) *	***0.047***	0.03(0.08)	0.137
BMI Zscore	2.9(0.4)	2.7(0.4)	***0.053***	−0.2(0.1)	3.1(0.8)	3.0(0.8)	***0.063***	−0.1(0.5)	0.074
Pli(mm)	141.2(29.8)	134.9(27.4) *	***0.033***	−5.5(12.0)	138.1(27.0)	124.2(30.3) *	***0.002***	−14.6(26.3) #	***0.002***

*p* = pre-post intra group; * = *p* < 0.05; delta = pre-post intra group diference; *p* delta = inter grup delta; # = *p* < 0.05 delta from the control group. (Bold and italic = statistically significant).

**Table 4 ijerph-17-05300-t004:** Data from the questionnaire on attitude toward physical education in girls and boys (Mean and SD)—pre-intervention.

Item	Girls	Boys	*p*
Subjet organization concordancy	15.2(3.8)	14.5(3.6)	0.373
PE Diifficulty	17.5(3.7)	16.7(3.3)	0.266
Emphaty towards teachers and subject	16.3(3.7)	13.9(2.9)	***0.001***
PE as sport	10.5(3.0)	9.9(2.6)	0.284
Preference for PE and Sport	10.5(2.6)	10.9(3.0)	0.499
Utility of PE	22.5(4.7)	22.3(4.5)	0.847
Value subject and PE Teacher	32.2(6.7)	29.9(6.7)	0.091

(Bold and italic = statistically significant).

**Table 5 ijerph-17-05300-t005:** Pre and post values of the study subjects on each dimension.

Item		Girls		Boys			
		Control	Experimental	Control	Experimental	*p*	*η* ^2^
Subjet organization concordancy	Pretest	14.2(4.1)	16.2(3.4)	12.8(3.3)	16.0(3.2)	0.536	0.04
	Postest	14.3(3.9)	14.4(3.8) ^*^	12.9(3.6)	13.5(3.7) ^*^		
PE Difficulty	Pretest	17.2(4.4)	17.8(3.0)	16.7(3.5)	16.8(3.2)	0.2	0.056
	Postest	17.1(3.9)	15.7(3.8) ^*^	17.0(3.7)	17.5(3.2)		
Emphaty towards teachers and subject	Pretest	16.7(3.9)	15.9(3.4)	14.3(2.2)	13.5(3.3)	***0.021***	0.055
	Postest	16.8(3.5)	14.3(3.9) ^*^	15.1(2.5) ^*^	15.0(342) ^*^		
PE as sport	Pretest	11.4(3.1)	9.7(2.7)	9.9(3.3)	10.0(1.8)	0.101	0.028
	Postest	11.3(2.7)	8.6(3.3) ^*^	9.7(2.6)	10.8(2.4)		
Preference for PE and Sport	Pretest	10.2(2.5)	10.8(2.8)	9.7(2.4)	12.0(3.0)	0.261	0.013
	Postest	10.0(2.2)	8.6(3.2) ^*^	9.6(2.4)	10.9(2.7)		
Utility of PE	Pretest	24.3(4.9)	20.7(3.)	21.1(4.4)	23.3(4.5)	0.315	0.011
	Postest	23.9(5.1)	20.7(4.0)	21.5(4.2)	24.3(5.3)		
Value subject and PE Teacher	Pretest	30.5(6.6)	33.9(6.5)	28.6(8.0)	31.0(5.2)	0.077	0.033
	Postest	31.3(6.4) ^*^	30.7(8.1)	29.3(3.2)	32.8(7.7)		

* *p* < 0.05 pre-post intra group; *p = p* value interaction gender × exercise; *η*^2^ = effect size of the interaction (Mean and SD). (Bold and italic = statistically significant).

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
