# Peer review of "Assessment of Attitudes Toward Physical Education by the Implementation of an Extracurricular Program for Obese Children"

_ijerph, 2020, doi:10.3390/ijerph17155300_

Round 1

Reviewer 1 Report

All of suggestions have been included.

Only must be change the value in table 3: 400m. baseline, in the table put 11.3, this value, the authors say that the correct value is 111.3

Author Response

Only must be change the value in table 3: 400m. baseline, in the table put 11.3, this value, the authors say that the correct value is 111.3

Its has been changed:

CG

EG

Baseline

Post

Delta

Baseline

Post

Delta

Curl Up(n)

8.1(3.9)

8.6(4.2)

0.5(2.3)

7.0(5.7)

8.5(5.9)*

2.0(0.3) #

Shoulder strech (cm)

4.9(4.0)

5.6(3.9)

0.7(1.5)

5.1(7.9)

6.1(6.9)*

1.2(2.7)

Push_up(s)

3.9(2.0)

5.2(1.7)*

1.4(1.5)

2.8(2.9)

3.5(2.3)*

0.6(2.1)

400m(s)

127(34.1)

129.8(3.9)

2.6(17.3)

111.3(26.5)

106.0(26.2)*

-5.4(12.8) #

Trunk_lift(cm)

32.7(5.2)

34.8(5.0)*

2.1(1.9)

32.8(6.3)

34.2(6.2)*

1.5(2.5)

H.jump feet together(m)

0.97(0.16)

0.99(0.17)*

0.02 (0.05)

0.97(0.17)

1.00(0.16)*

0.03(0.08)

BMI Zscore

2.9(0.4)

2.7(0.4)

-0.2(0.1)

3.1(0.8)

3.0(0.8)

-0.1(0.5)

Pli(mm)

141.2(29.8)

134.9(27.4)*

-5.5(12.0)

138.1(27.0)

124.2(30.3)*

-14.6(26.3) #

Reviewer 2 Report

Abstract 

Line 19-20: The sentence is not making sense. 

"Questionnaire of Attitudes towards Physical Education (CAEF) questionnaire was used" - was used to? 

Line 21-22: "The only difference..."- there's something wrong with this sentence as well

p=0,001 should be appearing as p=0.001 (use dot, not coma).

Materials & Methods

Line 88-89: The sentence is not making sense. 

Line 95 - 99: This part not making sense. Are the phrases in red font meant to be removed? I found this issue in Line 105, 133 as well!

Section 2.5 - Please rewrite this paragraph.

Results

Table 2 - check the title 

Line 192 - 200: I don't understand the coding H4, H5, H6 & H7 meant here.

Some empty lines between Line 243-244

Discussion

Move the section on limitations to Discussion. 

References

Please check the numbering

Author Response

Response to Reviewer 2:

First of all, we would like to express our sincere gratitude for all comments and suggestions received from the Reviewer 2. This information has certainly enriched the text for its best understanding, thank you very much indeed. We have clarified the reviewer’s questions. We have introduced the required changes both in our answers to the specific comments and in the final manuscript V2.

General comments:

Extensive editing of English language and style required

Response: Thank you very much for pointing it out. We have introduced the required changes. A professional English editor has reviewed the manuscript, which certificate is attached.

Abstract 

Line 19-20: The sentence is not making sense. 

"Questionnaire of Attitudes towards Physical Education (CAEF) questionnaire was used" - was used to? 

Line 21-22: "The only difference..."- there's something wrong with this sentence as well

p=0,001 should be appearing as p=0.001 (use dot, not coma).

Response: Thank you very much for pointing it out. We have introduced the required changes.:

 Abstract: The World Health Organization (WHO) identifies the importance of implementing physical activity programs such as physical education (PE) classes in schools. This study identifies the attitudes of obese children toward PE, before and after participation in a vigorous-intensity physical exercise program without the participation of normal-weight peers using a questionnaire on Attitudes toward Physical Education (CAEF). 98 children between 8–11 years of age were randomized in an Experimental Group (GE) (n = 48) and a Control Group (CG) (n = 47). They were assessed using a questionnaire on Attitudes toward Physical Education (CAEF). All the study participants exhibited a BMI Z-score ≥ 2.  Before the intervention, the only difference between boys and girls was "empathy to teacher and physical education subject" (p = 0.001, d de Cohen = 0.72, r = 0.34).. The interaction between gender and training was only present in empathy for the teacher, with a medium effect size (η2 =0.055). The implementation of PE with two hours per week elicits only a few effects over the attitude of obese children, even though with a certain engagement of gender through training in the adjustment of empathy for teachers and the PE class.

Materials & Methods

Line 88-89: The sentence is not making sense. 

Line 95 - 99: This part not making sense. Are the phrases in red font meant to be removed? I found this issue in Line 105, 133 as well!

Response: Thank you very much for pointing it out. We have introduced the required changes.:

Material and Methods

A total of 104 school children were grouped in an experimental and a control group separately, both with children with obesity. The Questionnaire for attitudes toward physical education (CAEF) was used to screen the study participants for a 20-week pre- and post-intervention period [29]. This questionnaire was applied in both groups  before and after of intevention

Section 2.5 - Please rewrite this paragraph.

Response: Thank you very much for pointing it out. We have introduced the required changes.:

2.6 Statistical analysis

Statistical analysis was performed with SPSS v.23.0 (SPSS Inc., IBM, USA). Data are expressed as mean ± standard deviation. The Kolmogorov–Smirnov test was performed to test the normality assumption. Student's independent t-tests were used to examine differences between groups and Student's paired t-tests for pre-post means within each group, and Cohen’s d was used to measure the effect size. The main outcome measure, the modification of attitude of obese boys and girls toward PE classes, was analyzed with the two-way ANOVA (sex and intervention group), interaction analysis partial eta squared (η2) as a measure of effect size. A level of significance was established at p<0.05.

Results

Table 2 - check the title 

Response: Thank you very much for pointing it out. We have introduced the required changes.:

Table 2.  Age and anthropometric characteristics of girls and boys in the sample (Mean and SD)

Line 192 - 200: I don't understand the coding H4, H5, H6 & H7 meant here.

Letters h appear as editor's notes, not from the content of the article.

Some empty lines between Line 243-244

Response: Thank you very much for pointing it out. We have introduced the required changes.

Discussion

Move the section on limitations to Discussion. 

Response: Thank you very much for pointing it out. We have introduced the required changes. 

References

Please check the numbering

Response: Thank you very much for pointing it out. We have introduced the required changes.

Round 2

Reviewer 2 Report

Overall, I am satisfied with the corrections that the authors have made in the manuscript.

There are some minor errors which may have been overlooked:

Table 2 - dot between numbers appearing as coma. mark the significant p value. 

Table 3 - provide the p values

Table 5 - p and  η2 definitions appeared as footnote but interac & effect were used in the table. mark the significant p value. 

Author Response

We sincerely appreciate the comments that have been made on several occasions, as they have significantly improved the article.

Sincerely, THANK YOU.

We have made the modifications to the reviewer's remarks:

Table 2 - dot between numbers appearing as coma. mark the significant p value.

We have replaced in the decimal separation "," by "." Values of p less than 0.05 have been highlighted in bold italics

Table 3 - provide the p values

We have completely replaced the table: in which we have added the pre-post intragroup p values and the p values of the inter-group delta comparison.

Values of p less than 0.05 have been highlighted in bold and italics.

We've changed the table's foornote, leaving it at that:

“p = pre-post intra group; * = p<0.05 ; delta = pre-post intra group diference; p delta = inter grup delta; # = p<0.05 delta from the control group.”

Table 5 - p and  η2 definitions appeared as footnote but interac & effect were used in the table. mark the significant p value.

We have corrected the error pointed out by the reviewer, replacing in the table "Interac" by "p" ; and "effect" by "η2".

Values of p less than 0.05 have been highlighted in bold and italics

The missing "*" from the pre-post intragroup comparison has been added

The effet size is a measure of the size of the effect, and its value has a qualitative interpretation, so it does not have a p.

This manuscript is a resubmission of an earlier submission. The following is a list of the peer review reports and author responses from that submission.

Round 1

Reviewer 1 Report

The work is interesting, it raises a very important issue. However, I have a few comments:

line 105: the title of Table 5 should be changed

line 157: the title of Table 2 should be changed

line 158 is: (Mean and DS)  - I suppose it should be (Mean and SD)  - this note applies to all work

line 161; have there been any differences in heart rate between girls and boys?

line 169: write the full name of the abbreviation CAEF in the title of the table 3

line 261: Author Contributions - this part is incomplete, some information is missing, e.g. who wrote the work, etc.

line 264: Acknowledgments - :” In this section you can acknowledge any support given which is not covered by the author 265 contribution or funding sections. This may include administrative and technical support, or donations in kind 266 (e.g., materials used for experiments).” this part should be deleted

Reviewer 2 Report

Abstract

1) Abstract has to be rewritten with correct use of English

2) line 21 -  the teacher and the subject of PE (p=0,001).

p=0.001? It will be more informative for readers if authors include the effect size

3) line 22- with a weak effect size (0,055).

what does the 0,055 indicates?

Introduction

Require language editing work. 

Methods

1) line 87 - Which were evaluated with the Questionnaire for attitudes towards physical education (CAEF) [29].

Unclear why this sentence appeared here.

2) line 86-90: I don't think this entire paragraph is making any sense. Usually we provide readers an overview of the study design at the start of this section. 

3) section 2.1: this can be improved by revamping the parts with this suggested flow: who were they? where they came from? who recruited them? how they were recruited? inclusion & exclusion criteria? and so on. 

4) Table 1 not serving much purpose here. 

Results

Poorly written for readers to understand the flow or the presentation. 

Discussion

What are the study strengths and limitation? Move some of these discussion that can be seen in Conclusion. 

Conclusion

Summarise the findings, recommend some implications on practice and take home message for future studies in this area. 

Reviewer 3 Report

GENERAL COMMENTS

The aim of the current study was to identify the attitude of obese children towards Physical Education before and after participating in a vigorous-intensity physical exercise program. The experimental study included 98 children (48 from the experimental group and 47 from the control group), all children were classified as obese (with BMI Z-Score ≥2). The topic is relevant. Despite strengths, the article has also some weak points (corrections to text editing and some important methodological aspects were not reported in the text) which need to be improved in order to be published. Please see the specific comments.

SPECIFIC COMMENTS

Abstract

The abstract should be but without headings.

Some of the abbreviations appeared in the abstract the first time (WHO, EG, CG, CAEF) – all of them should be defined in parentheses. Please, add the full name of them.

It will be great to add the age of the children in the abstract.

Introduction

Line 50: Please, add an abbreviation for physical activity (PA) and use it later in the text.

Line 54: Please, use only an abbreviation for Physical Education (PE) in the main text.

Based on the aim, did the Authors generate any official hypotheses for this work? If so, it will be helpful to include in the introduction.

Method

Line 99: There is relevant information on the sample that was not described. They should be included in the text: How many schools were included in the study? How were they selected? How many children were invited? Were there refusals and losses?

Line 109: Why the program took 20 weeks, 2 times per week. A justification should be provided for the selection of the duration of the program.

Line 115-116: Only children with obesity participated in the activities of this program. How did you explain to those children why they were selected for this program?

Line 130: Was the Cronbach's alpha calculated for total items of the Questionnaire for attitudes towards Physical Education (CAEF)? Is it a questionnaire valid and reliable also to investigate attitudes toward PE in Mexican children? This aspect is very important to support your results.

Discussion

Please, indicate some limitations and strengths of the study and also to include mention of the originality and value of this study.

Please, try to translate references in the Spanish language to English where possible.

Reviewer 4 Report

As authors state, the increase of obesity in school children may be taken into account by PE professionals for designing their pedagogic approach in order to encourage children to acquire healthy habits for the rest of their lives. In my opinion, this is an interesting field of research and this work is worthy.

However, there are some points that should be addressed:

- Line 113: Authors mention that children performed a physical activity at 79.8 of their maximum heart rate, but the criteria for establishing these maximum heart rates is not explained. In the same way, one of limitations that may be listed in the text is the problems associated with the use of heart frequency as a parameter for effort control (Strath et al., 2013)

- Line 174: In the legend of table 4, authors state that one and two asterisks are used for highlighting significant results. However, they are not included next to the p<0.05 or p<0.01.

- Line 188: The sentence "Their values are interpreted in reverse, therefore,..." refers to the interpretation of the scores obtained. It would be worth to indicate which values denote a better result with a higher score and which values denote a better result with a lower score, in order to facilitate data interpretation by the reader. A sentence may be placed explaining that higher values indicate a better result with the exception of those items where lower values indicate a better result.

- Line 202: The acronym EG may be presented in line 88, where the term is introduced in the text.

- Line 205: In table 5, the EG value in Baseline of 400m (s) is 11.3 (26.5). Review that the value is correct, and correct it if it's a data error

It is evident that physical condition parameters of children with obesity will be improved after performing two sessions per week during 20 weeks. Although it was not one of the study’s objectives, it would be interesting to include in future research lines if this physical condition improvement benefits the self-concept perception of these children, and also if there is a change in the vision-acceptance of physical activity (in general) and physical education (in particular) as an element to incorporate into their lifestyle by these children.

Strath, S. J., Kaminsky, L. A., Ainsworth, B. E., Ekelund, U., Freedson, P. S., Gary, R. A., Richardson, C. R., Smith, D. T., Swartz, A. M., & on behalf of the American Heart Association Physical Activity Committee of the Council on Lifestyle and Cardiometabolic Health and Cardiovascular, E. (2013). Guide to the Assessment of Physical Activity: Clinical and Research Applications A Scientific Statement From the American Heart Association. Circulation, 01.cir.0000435708.67487.da. https://doi.org/10.1161/01.cir.0000435708.67487.da